# HIV and COVID-19 Co-Infection: Epidemiology, Clinical Characteristics, and Treatment

**DOI:** 10.3390/v15020577

**Published:** 2023-02-20

**Authors:** Dimitris Basoulis, Elpida Mastrogianni, Pantazis-Michail Voutsinas, Mina Psichogiou

**Affiliations:** 1COVID-19 Department, Laiko General Hospital, National and Kapodistrian University of Athens, 11527 Athens, Greece; 2First Department of Internal Medicine, Laiko General Hospital, National and Kapodistrian University of Athens, 11527 Athens, Greece

**Keywords:** HIV, COVID-19, epidemiology, clinical characteristics, treatment

## Abstract

The COVID-19 pandemic has been a global medical emergency with a significant socio-economic impact. People with HIV (PWH), due to the underlying immunosuppression and the particularities of HIV stigma, are considered a vulnerable population at high risk. In this review, we report what is currently known in the available literature with regards to the clinical implications of the overlap of the two epidemics. PWH share the same risk factors for severe COVID-19 as the general population (age, comorbidities), but virological and immunological status also plays an important role. Clinical presentation does not differ significantly, but there are some opportunistic infections that can mimic or co-exist with COVID-19. PWH should be prime candidates for preventative COVID-19 treatments when they are available, but in the setting of resistant strains, this might be not easy. When considering small-molecule medications, physicians need to always remember to address potential interactions with ART, and when considering immunosuppressants, they need to be aware of potential risks for opportunistic infections. COVID-19 shares similarities with HIV in how the public perceives patients—with fear of the unknown and prejudice. There are opportunities for HIV treatment hidden in COVID-19 research with the leaps gained in both monoclonal antibody and vaccine development.

## 1. Introduction

The COVID-19 pandemic poses a significant global health threat, with a large socio-economic burden [1]. As of 3 February 2023, 754,018,841 confirmed cases of COVID-19 and 6,817,478 deaths have been reported globally [2].

At the same time, 38.4 million people globally are living with HIV, of whom approximately 75% have access to antiretroviral treatment (ART). Almost 1.5 million people became newly infected with HIV in 2021, while 85% of all people with HIV (PWH) were aware of their HIV status in 2021 [3]. HIV prevention and control programs have been largely disrupted by the COVID-19 pandemic, especially in low- and middle-income countries where HIV referrals for diagnosis and treatment as well as HIV testing dropped by 37% and 41%, respectively, between April and September 2020, compared with the same period in 2019 [4].

Age is the most influential risk factor for severe SARS-CoV-2 infection and poor disease progression. Other underlying comorbidities, such as obesity, diabetes, hypertension, cardiovascular disease, heart failure, chronic kidney disease, and malignancy, further increase the risk of severe infection and mortality due to COVID-19 [5,6,7]. Race and ethnicity are additional important factors, with some racial and ethnic minority groups being at higher risk for adverse outcomes [8]. The Centers for Disease Control and Prevention currently identifies PWH as being at elevated risk for severe illness from COVID-19, especially if they are older, carry underlying medical conditions, have advanced HIV disease, or are not on ART [9].

However, the interaction between SARS-CoV-2 and HIV infection is still not completely clear. HIV infection is characterized by immune dysregulation. Chronic interferon signaling, T-cell exhaustion and defective B-cell function with polyclonal yet ineffective antibody production might render PWH more vulnerable to severe SARS-CoV-2 infection [10]. Long before the advent of COVID-19, HIV had been associated with a higher risk of severe outcomes from other respiratory infections [11]. The state of immunosuppression largely is dependent on the receipt of ART, with people not on ART being more susceptible to opportunistic infections [12]. However, not all defects are entirely reversed after ART initiation, and even when viral load is not detectable, PWH continue to demonstrate suboptimal control of viral infections [13]. Moreover, non-communicable diseases such as cardiovascular and metabolic comorbidities or malignancy are more prevalent in PWH, compared to people without HIV (PWoH), thus adding risk factors for COVID-19 severity [14].

Our knowledge of the interplay between HIV and SARS-CoV-2 infection is further challenged by the likely emergence of the Omicron variant in southern Africa and its possible link to the HIV pandemic, which is a main cause of immunosuppression in the region. This theory is based on the concept that a prolonged infection in an immunocompromised host might have given rise to the Omicron variant of SARS-CoV-2 [15], as it has already been described for other multinational SARS-CoV-2 variants [16].

## 2. Epidemiology

According to meta-analysis data, the global pooled prevalence of PWH among COVID-19 cases is estimated to be 2%. At the continental level, the pooled prevalence for Europe, North America, Africa and Asia was 0.5%, 1.2%, 11% and 1%, respectively. The high rates in North America can be attributed to the fact that studies from the USA were represented mainly by studies from the states of New York and Georgia, which are known for higher HIV positivity rates in the population. The highest prevalence from Africa is also explained by the contribution of data from eastern and southern Africa, regions highly affected by the HIV pandemic [17]. These results are also confirmed by another meta-analysis, which showed a pooled global prevalence of 26.9‰, while the pooled prevalence in Africa was 118.5% [18]. An intersection between new HIV infection and COVID-19 has been documented in the USA with a spatial correlation of 0.39, which was not affected by demographic, social, economic or behavioral characteristics. Furthermore, income inequality was associated with a higher prevalence of both diseases [19].

Other large clinical cohorts have shown that although PWH are more likely to be tested for SARS-CoV-2, there is no evidence of higher positivity rates compared to PWoH [20,21]. However, prolonged viral shedding might be detected in the setting of prolonged HIV infection, low CD4 counts and unsuppressed viral load, which has previously been described in immunocompromised hosts [22,23]. Regarding the likelihood of hospitalization due to COVID-19 among PWH, rates of hospital admission range from 0.7% to 1.9% [24]. While findings from case series and larger studies did not suggest increased rates of hospital admission in this patient population [25,26], results from a meta-analysis of six studies showed that PWH are indeed more likely to be hospitalized due to COVID-19 compared to PWoH (OR: 1.49; 95% CI 1.01–2.21) [18].

## 3. Clinical Outcomes

To date, data regarding the clinical outcomes of COVID-19 in PWH have been conflicting, with discordant results between different regions and within specific geographical territories. Early in the course of the pandemic, single-center studies based on small cohorts of patients showed that PWH had a similar [27,28,29] or even lower [30] risk of severe disease and mortality, compared with PWoH. Conversely, collective data from several centers in the UK showed that PWH had a 2.9 times higher risk of COVID-19 death than PWoH after adjusting for age and gender. Interestingly, among PWH, the risk was significantly greater for people of black ethnicity compared to non-black ethnicity, with a hazard ratio (HR) of 4.31 versus 1.84 [31]. A study conducted in New York State showed an increased risk of ICU admission and intubation among PWH, but only in the group younger than 50 years, with no significantly increased risk in other age groups [32]. Another study in New York showed that HIV infection alone is associated with an increased risk for severe disease and hospitalization, while the risk of hospitalization was even higher among those with advanced HIV disease [29,33]. Furthermore, meta-analyses have had inconsistent findings, with some showing increased odds of mortality among PWH [34,35,36,37] and others showing no differences in outcomes between HIV and non-HIV populations [38].

The largest international multicenter study on hospitalized patients with COVID-19 to date, with data collected by the WHO Global Clinical Platform for COVID-19, came to the conclusion that PWH were 15% more likely to develop severe or critical COVID-19 and were 38% more likely to die in hospital compared to PWoH [39]. Among PWH aged 45–75 years, male gender, chronic cardiac disease, or hypertension were risk factors for severe COVID-19. Moreover, for ages older than 18 years, being male or having diabetes, hypertension, malignancy, tuberculosis, and chronic kidney disease were risk factors for death in hospital. The inclusion of hospitalized patients only made the study unable to define risk factors leading to hospitalization. Caution is advised in the extrapolation of findings from this and other African studies with similar results, since their findings might not be applicable in western countries [39,40]. Data from the largest cohort of US COVID-19 cases showed that PWH had a higher prevalence of all comorbidities compared to PWoH [41].

HIV opportunistic infections and related co-infections have also been noted as potential risk factors for adverse outcomes. Tuberculosis co-infection was associated with an increased likelihood of COVID-19 death [40,42]. In a systematic review of case reports and case series, it appeared that HCV co-infection is major risk factor of hospital admission and critical condition [43].

Remarkably, most of our knowledge relies on studies conducted in the pre-vaccination era or data from low- and middle-income countries, where vaccination coverage was low. More studies with data on vaccination status will shed light on the impact that HIV infection has on COVID-19 outcomes. As an example, fully vaccinated PWH were not found to be at increased risk of severe illness compared to PWoH. Nevertheless, the risk of severe breakthrough illness was 59% higher in PWH with a CD4 cell count < 350 cells/μL compared to PWoH. Among PWH, previous COVID-19 seemed to be protective [44].

The association of poor outcomes and low CD4 counts has been verified by several studies from the US [45,46]. A multicenter study proved that viremia was significantly associated with COVID-19 disease severity and that CD4 a T-cell count < 200 cells/mm^3^ is not only a factor that leads to increased mortality but also to an increased probability of admission to hospital [47]. Although data from the US National COVID Cohort Collaborative showed that a lower CD4 cell count among PWH is associated with a higher risk of adverse COVID-19 outcomes, PWH without virological suppression had an increased risk of hospitalization but not death [41]. Among PWH, unsuppressed viral load has been consistently associated with poor COVID-19 outcomes [39,48].

Data on the impact of ART on the clinical course of COVID-19 among PWH are mixed and limited. PWH under ART had a less severe clinical presentation of COVID-19 than the general population regarding better prognosis and faster resolution of symptoms [39]. On the other hand, there are cohorts which did not show that ART provided protection against COVID-19 severity [27]. A limited multicenter case series study from Spain has shown that the type of ART does not influence the outcome of COVID-19 [49]. On the contrary, data from a multicenter cohort from Madrid HIV clinics with more than 75,000 participants showed that the risk for COVID-19 hospitalization differed based on backbone regimens, with the risk per 10,000 persons being calculated at 10.5 among those receiving TDF/FTC, 20.3 among patients receiving TAF/FTC, 23.4 among those receiving ABC/3TC, and 20.0 for those receiving other regimens [50].

Recent data from South Africa show the association between COVID-19 variants and mortality in PWH. During the Alpha and Beta variant waves of the pandemic, mortality rates among hospitalized patients were 24% for PWH and 21% for PWoH. Death rates did not change significantly through 2021 when the Alpha, Beta and Delta variants were circulating. In 2022, with the predominance of the Omicron variant, while the death rate in HIV-negative people fell to 8%, the death rate for HIV-positive people remained high at 19.8%. This continuing high death rate in PWH might be explained by the low vaccination coverage in South Africa, where only 32% of the population is fully vaccinated against COVID-19 [51]. These findings underscore the importance of prioritizing PWH for vaccination, and indeed, recent studies show that PWH are able to mount a strong humoral response to COVID-19 vaccines, especially if they are on ART with suppressed viral loads, higher CD4 cell counts, and higher CD4/CD8 ratios [52]. Even under the condition of low CD4 counts, despite low humoral responses, the CD4 cellular immune response to vaccination seems to be preserved [53].

Apart from medical risk factors, socioeconomic factors play a role in COVID-19 outcomes among PWH. Racial disparities in prevalence and outcomes for HIV disease are well described, and a similar pattern of incidence and outcome appears to follow the COVID-19 disease [54]. Data indicating poor outcomes among Black PWH and those living in high-poverty neighborhoods come from USA [55], UK [31] and Paris [56].

## 4. Clinical Characteristics

Many large cohort studies have reported clinical characteristics of COVID-19 in the general population, but data on COVID-19 among PWH are relatively scarce.

In clinical data derived from the WHO Global Clinical Platform between 1 January 2020 and 1 July 2021, the mean age of PWH with COVID-19 was 45.5 years. They were more likely to be younger and female than PWoH [39]. The three most frequent symptoms were cough (62.1%), fever (55.7%), and shortness of breath (51.9%) [39]. According to a multicenter study of 286 PWH in which the mean age of patients was 51.4 years, 25.9% were female, 88.7% had HIV virologic suppression, 94.3% were on antiretroviral therapy, and 80.8% had comorbidities; the most common presenting symptoms of COVID-19 among people with HIV were cough (76.2%), fever (70.7%), and fatigue (66%) which are similar to the general population. Moreover, symptoms such as fever, fatigue, dyspnea, gastrointestinal discomfort and altered mental status were more common among PWH who were hospitalized, while upper respiratory symptoms such as sore throat, nasal congestion and headache were seen in PWH who were not hospitalized [45]. These results echo the results of smaller studies from the USA [57], United Kingdom [58], Germany [59] and Italy [60].

In the Multicenter AIDS Cohort Study and the Women’s Interagency HIV Study (MACS/WIHS) Combined Cohort Study (MWCCS), where the median age of the population was 57 years, 74% had undetectable HIV viral loads, and the median CD4+ T lymphocyte cell count was 682 cells/mm^3^, symptom profiles were similar in PWH and HIV seronegative participants, including headache (23% vs. 24%), myalgias (19% vs. 18%), shortness of breath (14% vs. 13%), chills (12% vs. 10%), fever (6% vs. 6%) and loss of taste or smell (6% vs. 7%), apart from rhinorrhea, sore throat and cough, for which prevalence was slightly but significantly lower in PWH [61]. In a review from China, 9.6% of participants were asymptomatic, 44.2% had mild and 46.2% severe COVID-19 infection. More specifically, 41.9% reported respiratory distress, 41.9% fatigue, and 26.7% gastrointestinal symptoms [62].

The majority of the literature focuses on virologically suppressed PWH, but there are cases with poorly controlled HIV or acquired immune deficiency syndrome (AIDS) and opportunistic infections at the same time. Of special interest is co-infection in PWH with COVID-19 and *Pneumocystis jirovecii* pneumonia (PJP) since the two infections may present with similar features of exercise desaturation, dry cough and relatively normal chest auscultation on clinical examination [63,64]. Also, there has been a case of a person with previously undiagnosed HIV infection who presented with COVID-19, PJP and cytomegalovirus pneumonitis simultaneously. His course was further complicated with immune reconstitution inflammatory syndrome (IRIS) [65]. Another particularly interesting case involved a patient with COVID-19 and AIDS-related disseminated histoplasmosis [66].

Finally, the data available on risk factors for post-acute COVID-19 syndrome among PWH are limited. A retrospective observational cohort study from the US showed that PWH, when adjusting for demographics, comorbidities, and severity of illness, were significantly more likely to report persistent symptoms at least nine months out from diagnosis [67]. A case-control study in the pre-vaccine era found that PWH had 4.01-fold higher odds of post-acute COVID-19 syndrome when adjusting for age, hospitalization and time since infection [68].

## 5. Management of COVID-19 in PWH

There exists a consensus amongst HIV societies with regards to the treatment of PWH and COVID-19 co-infection. BHIVA, IDSA, EACS, NIH, IAS and DHHS guidelines recommend a standard treatment for COVID-19 in HIV infected individuals [69,70,71,72,73,74]. The current guidelines are summarized in Figure 1.

The topic of vaccination is beyond the scope of this review, yet it should be noted that all guidelines recommend that PWH be vaccinated fully and with booster doses against SARS-CoV-2. Several studies have shown that the immunogenicity of available vaccines, both in the main vaccination and the booster vaccination, is similar in healthy controls and PWH, especially when CD4 counts are well preserved [75,76,77,78]. CD4 counts <250 cells/mL have been associated with decreased immunogenicity, and it appears that humoral immunity conferred through vaccination wanes faster in PWH, making booster vaccination doses more important. Finally, the response to different types of vaccines was reported in a meta-analysis to be similar in PWH [79].

### 5.1. Pre-Exposure Prophylaxis

Physicians caring for PWH have long been accustomed to the idea of pre-exposure prophylaxis in the case of HIV infection. The advent of AZD7442, a monoclonal antibody combination of tixagevimab and cilgavimab—both neutralizing antibodies against SARS-CoV-2—was a natural evolution of other monoclonal antibodies that had been tried for the treatment of COVID-19, but in this case, due to the extended half-life of the antibodies used, the combination was viable as pre-exposure prophylaxis [80].

In the PROVENT trial, 5197 participants were randomized 2:1 to receive either AZD7442 or a placebo and were monitored for the development of a new COVID-19 infection. Primary analysis was planned after 24 primary end-point events had been confirmed or 30% of the participants had been unblinded. The median follow-up time for the primary analysis was 83 days, at which time symptomatic SARS-CoV-2-PCR-positive illness had occurred in 8 of 3441 participants (0.2%) in the AZD7442 group and in 17 of 1731 participants (1.0%) in the placebo group. This difference was statistically significant, with a relative risk reduction of 76.7%. [80] However, only a very small fraction of HIV patients participated in the trial, with the authors listing only 15 (0.4%) and 9 (0.5%) participants with “immunosuppressive disease” (without further clarification if any of those were PWH at all) in the AZD7442 and placebo arms, respectively [80]. Nonetheless, the monoclonal combination has been approved for use in PWH to prevent COVID-19 infection. Tixagevimab/cilgavimab at a dose of 150/150 mg is recommended for individuals with advanced or untreated HIV infection, defined as having CD4 counts < 200 cells/mm^3^ or a history of an AIDS-defining condition without immune reconstitution or clinical manifestations of symptomatic HIV [81].

Tixagevimab/cilgavimab is not associated with significant adverse events, aside from topical injection site reactions and hypersensitivity. There have been reports of myocardial infarction and new onset heart failure, but no clear associations have been established [81].

What has changed since the approval of the combination antibody is the emergence of the various Omicron variant lineages. It soon became apparent from various studies that the ability of the AZD7442 combination to neutralize the Omicron variants was diminished. In one of the first reports, Aggarwal et al. showed that cilgavimab had completely lost its effectiveness against B.1.1529, while tixagevimab had a 73.8-fold reduction compared to the A.2.2 ancestral strain [82]. Zhou et al. reported a 359-fold activity decrease against BA.1 and 1920-fold against BA.2 compared to the D614G ancestral virus [83]. In a recent metanalysis of several studies, the combination showed a median 86-fold effectiveness reduction against BA.1 and a median 5.4-fold reduction against BA.2 [84]. All this data led the FDA authorities to recommend using double the approved dose for COVID-19 pre-exposure prophylaxis [85].

With regards to the BA.5 lineage, it seems that the monoclonal combination retains some activity. Touret et al. showed that even though tixagevimab alone is completely ineffective against BA.1, BA.2 and BA.5 compared to the B.1 ancestral strains, the cilgavimab component has a 84.2-fold, 9.6-fold and 18.7-fold reduced neutralization effectiveness, and the combination has a 29.4-fold, 1.9-fold and 2.8-fold reduced effectiveness, respectively; thus, it remains a viable option for prevention [86]. Even more recently, the emergence of newer Omicron sub-lineages has put into question the effectiveness of AZD7442 for the future. Planas et al. have shown that cilgavimab and tixagevimab alone and in combination have lost all effectiveness against the newly emerging subvariants BA.2.75.2 and BQ.1.1 [87]. It should be noted, however, that information pertaining to the decreased effectiveness of this combination against the omicron lineages is based on in vitro data, and how this would translate to real-world in vivo data is not entirely certain. Nonetheless, the IDSA has issued a recommendation in January 2023 against the use of tixagevimab/cilgavimab in the US, since at that time, the dominant variants displayed resistance to neutralization, but it maintained that in regions of the world where the dominant variants were susceptible, the monoclonal antibody combination was still a viable choice. [88]

### 5.2. Post-Exposure Prophylaxis

The only treatment that had been included in the guidelines for post-exposure prophylaxis had so far been the monoclonal antibody combination of casirivimab/imdevimab. Both antibodies target non-overlapping epitopes of the SARS-CoV-2 spike protein receptor binding domain [89]. The justification for this use derived from a double-blind trial before the advent of the Omicron variants, in which household contacts of verified index patients were randomized to receive the monoclonal antibody combination or a placebo via subcutaneous injection and assessed for COVID-19 infection at 28 days. The primary analysis cohort included 1505 participants randomized 1:1. Symptomatic SARS-CoV-2 infection developed in 11 of the 753 participants in the antibody combination group (1.5%) and in 59 of the 752 participants in the placebo group (7.8%) (relative risk reduction, 81.4%, OR 0.17, *p* < 0.001) [89]. No PWH participated in the trial.

With the emergence of Omicron, however, the usefulness of this approach was put to the test because it was rapidly demonstrated that both components of this combination were lacking in effectiveness against the new variant. Several reports have shown that casirivimab/imdevimab has no activity against BA.1, and that casirivimab alone has no activity against BA.2, whereas imdevimab retained its effectiveness, albeit several hundred-fold reduced [84,90,91]. Against BA.5, casirivimab displays no activity, and imdevimab has been shown to also have reduced activity. Finally, the combination is ineffective against the emerging BQ1.1 and BA.2.75.2 strains [87]. Thus, the use of casirivimab/imdevimab as a post-exposure prophylaxis is limited by the current epidemiological data.

### 5.3. Early Outpatient Treatment

#### 5.3.1. Small Molecules

For individuals that have contracted SARS-CoV-2 but remain asymptomatic or with minimal symptoms and do not require supplemental oxygen, three medications have been approved for use for the prevention of deterioration and in order to avoid needing hospitalization: Molnupiravir, nirmatrelvir/ritonavir, and the 3-day remdesivir regimen. There are no specific trials addressing the effectiveness of these medications in PWH, but the recommendations for treatment remain the same as in the general population.

Molnupiravir is a small-molecule ribonucleoside prodrug of N-hydroxycytidine (NHC) with activity against SARS-CoV-2. NHC is transported intracellularly and phosphorylated into triphosphate NHC which is then incorporated into viral RNA by the viral RNA polymerase. Subsequently, the viral polymerase incorporates adenosine or guanosine because it misreads NHC, leading to the accumulation of a progressively increasing numbers of errors, rendering the virus unable to replicate further [92]. In the MOVe-OUT trial, 1433 non-hospitalized SARS-CoV-2-PCR-positive participants were randomized 1:1 to receive either molnupiravir or a placebo for the prevention of hospitalization. At 29 days post-randomization, 7.3% vs. 14.1% (difference −6.8, 95%CI −11.3 to −2.4, *p* = 0.001) required hospitalization or had died. Overall, molnupiravir showed a 30% reduction in the composite endpoint of hospitalization or death. No individuals with HIV infection participated in the trial, and patients with advanced disease were excluded, in accordance with protocol [93]. Molnupiravir is not associated with significant adverse events, aside from gastrointestinal discomfort, and it is not expected to have clinically significant drug–drug interactions. However, there is a significant pill burden associated with molnupiravir use (four capsules twice daily), albeit for 5 days only, that might deter PWH who might be already receiving older ART multi-pill regimens. There exists a small concern with regards to the ability of molnupiravir to cause host-cell mutagenesis, and thus, it should be avoided in pregnant women and small children [94].

Nirmatrelvir is a small-molecule antiviral agent targeting the SARS-CoV-2 3-chymotrypsin–like cysteine protease enzyme (Mpro) [95]. By inhibiting viral polyprotein transformation into functional units, it essentially inhibits viral replication. Ritonavir is a CYP3A4 inhibitor well known to physicians treating PWH since it has been used as part of treatment in individuals receiving protease inhibitors. Nirmatrelvir is metabolized by CYP3A4, and the inclusion of ritonavir in this regimen enhances the pharmacokinetics of nirmatrelvir, increasing its bioavailability [95]. Nirmatrelvir/ritonavir was approved as a treatment for the prevention of COVID-19 deterioration in high-risk individuals based on the findings of the EPIC-HR trial. A total of 2246 participants were randomized 1:1 to receive nirmatrelvir/ritonavir or a placebo. The incidence of COVID-19–related hospitalization or death by day 28 was lower in the nirmatrelvir group than in the placebo group by 6.32 percentage points (95% CI, −9.04 to −3.59; *p* < 0.001). The relative risk reduction was calculated at 89.1% [96]. Only one HIV-positive individual participated in the trial. Nirmatrelvir/ritonavir is not expected to cause any significant adverse events aside from gastrointestinal discomfort, but it bears significant drug–drug interactions that need to be accounted for when being prescribed, and it is contraindicated in patients with eGFR < 30 mL/min. With regards to potential interactions with ART, regimens containing ritonavir or cobicistat-boosted medications will exhibit potentially significant interactions that warrant closer monitoring for adverse events. It is however recommended that ART regimens be continued without modifications [74]. Nirmatrelvir/ritonavir also carries a significant pill burden (two tablets of nirmatrelvir and one tablet of ritonavir twice daily), again only for 5 days.

Remdesivir is a direct-acting nucleotide prodrug inhibitor of the SARS-CoV-2 RNA-dependent RNA polymerase that has been approved for use as a specific antiviral treatment for COVID [97]. After its initial FDA and EMA approval for use in hospitalized patients requiring oxygen, the PINETREE trial demonstrated that an early 3-day regimen of remdesivir can be used in high-risk individuals to prevent disease progression. A total of 562 patients were randomized 1:1 to receive intravenous remdesivir or a placebo. COVID-19-related hospitalization or death from any cause occurred in two patients (0.7%) in the remdesivir group and in fifteen (5.3%) in the placebo group (HR 0.13; 95% CI 0.03–0.59, *p* = 0.008) [98]. Only a few immunocompromised patients participated in the trial, and investigators did not disclose if any PWH participated. Remdesivir is not associated with significant adverse events aside from transient increases in transaminase levels nor with any important drug–drug interactions, but the intravenous mode of treatment might deter some patients, and it poses potential logistical problems.

Real-world data have shown great variance with regards to the effectiveness of these treatments. In a multicenter Japanese study during the Omicron surge, molnupiravir was shown to reduce the risk of deterioration by 55%, independent of other factors such as age or comorbidities [99]. In the British observational cohort OpenSAFELY study, the investigators attempted to compare effectiveness between molnupiravir and sotrovimab (a monoclonal neutralizing antibody) during a period covering both the Delta and Omicron variants. In their primary analysis cohort, they included 6020 patients (3331 in the sotrovimab arm and 2689 in the molnupiravir, with 73 and 118 PWH in each arm). Only a small percentage of patients met the primary endpoint of hospitalization or death; 2.05% in the molnupiravir group and 0.96% in the sotrovimab group, comparing the two, showed that sotrovimab was associated with a substantially lower risk compared to treatment with molnupiravir (HR 0.54, 95% CI 0.33 to 0.88, *p* = 0.01) [100]. In the AGILE CST-2 trial in the UK, molnupiravir was compared to placebo in vaccinated adults (since the original MOVe-OUT trial was conducted on unvaccinated adults solely). In a Bayesian analysis, molnupiravir did not meet superiority criteria vs. placebo for effectiveness in either vaccinated or unvaccinated groups, although participants in the molnupiravir arm would yield negative PCRs three days earlier compared to the controls [101].

In Israel Saliba et al., a propensity score matched analysis of 2261 molnupiravir-treated individuals compared to 2261 controls found that although molnupiravir was not associated with a reduced risk of disease progression (HR 0.83, 95% CI 0.57–1.21), there was a benefit amongst older people and those with inadequate vaccination [102]. Another large study from the same group investigated the effectiveness of nirmatrelvir/ritonavir in vaccinated and unvaccinated participants. Both nirmatrelvir/ritonavir and adequate COVID-19 vaccination status were associated with a significant decrease in the rate of severe COVID-19 or mortality, with adjusted HRs of 0.54 (95% CI 0.39–0.75) and 0.20 (95% CI 0.17–0.22). The protective effect of nirmatrelvir/ritonavir was the same in both vaccinated and unvaccinated individuals [103]. Similarly, in a retrospective study in the US of 756,036 individuals, nirmatrelvir/ritonavir use reduced the probability of hospitalization by 50% in booster-vaccinated, vaccinated and unvaccinated adult [104].

Several population studies from Hong Kong compared these treatments. Among 93,883 participants (5808 on nirmatrelvir/ritonavir, 4921 on molnupiravir and 83,154 controls) and after propensity score weighting, nirmatrelvir/ritonavir use (HR 0.79, 95% CI 0.65–0.95, *p* = 0.011) but not molnupiravir use (HR 1.17, 95% CI 0.99–1.39, *p* = 0.062) was associated with a reduced risk of hospitalization [105]. In another study, the authors compared molnupiravir to nirmatrelvir/ritonavir using inverse probability of treatment weighting analysis and estimated that in a cohort of roughly 30,000 participants, both molnupiravir (HR 0.31, 95% CI 0.24–0.40, *p* < 0.0001) and nirmatrelvir/ritonavir (HR 0.10, 95% CI 0.05–0.21, *p* < 0.0001) were significantly associated with reduced mortality. Only eleven PWH participated in this cohort [106].

A single-center Italian study from Pisa found no significant difference in effectiveness when comparing molnupiravir (98.2%) to nirmatrelvir/ritonavir (99.2%) or 3-day remdesivir (94.9%), although nirmatrelvir/ritonavir proved more efficient at preventing the need for hospitalization when compared to remdesivir [107].

There have been several reports both in the media [108] and in the scientific literature [109,110,111] with regards to the rebound phenomenon experienced in a minority of patients after treatment with nirmatrelvir/ritonavir, mainly and less frequently with molnupiravir. In the largest available cohort based on population electronic health records in the US, including roughly 11,000 patients treated with nirmatrelvir/ritonavir and 2300 with molnupiravir, the 7-day and 30-day COVID-19 rebound rates after nirmatrelvir/ritonavir treatment were 3.53% and 5.40% for COVID-19 infection, 2.31% and 5.87% for COVID-19 symptoms, and 0.44% and 0.77% for hospitalizations, while the 7-day and 30-day COVID-19 rebound rates after molnupiravir treatment were 5.86% and 8.59% for COVID-19 infection, 3.75% and 8.21% for COVID-19 symptoms, and 0.84% and 1.39% for hospitalizations [112]. The authors performed a propensity score match to compare rebound odds between the two treatments and found no differences. Rebound phenomena have not been associated with disease progression, and currently, most experts advise watchful waiting instead of re-treating patients [113].

#### 5.3.2. Monoclonal Antibodies

The use of monoclonal antibodies for COVID-19 treatment has been plagued by the emergence of new variants exhibiting resistance to available medications. Thus far, the combinations of bamlanivimab/etesevimab and casirivimab/imdevimab have been withdrawn from use due to ineffectiveness against the various Omicron lineages.

Sotrovimab was a promising engineered human monoclonal antibody with the ability to neutralize not only SARS-CoV-2, but also other sarbecoviruses. It was hypothesized that a monoclonal antibody that neutralizes all sarbecoviruses would target a highly conserved epitope that would be functionally retained as SARS-CoV-2 evolves [114]. It was approved for use based on the COMET-ICE trial, wherein 583 patients were randomized 1:1 to receive sotrovimab or a placebo. A total of 3 of the 291 patients in the sotrovimab group (1%), as compared with 21 of the 292 patients in the placebo group (7%), had disease progression leading to hospitalization or death (relative risk reduction, 85%). In vitro studies, however, demonstrated that sotrovimab had lost activity against the BA.2 strain almost completely [90]. Even worse, it has been demonstrated that in immunocompromised patients undergoing treatment with sotrovimab, treatment failure was associated with emerging spike mutations conferring resistance to sotrovimab [115]. On the other hand, two case reports suggested that the use of sotrovimab as an adjunctive to remdesivir therapy in severely immunocompromised AIDS patients showed a dramatic improvement in symptoms and rapid viral clearance [116,117].

Bebtelovimab is the newest monoclonal antibody that received FDA approval for use. Bebtelovimab binds to an epitope that is largely distinct from the mutations identified to be widely circulating within the newly emerged variants, including mutations that reduce the effectiveness of vaccines [118]. At the time of its approval, it was able to potently neutralize all circulating variants of concern, including BA.1 and BA.2. Access to bebtelovimab has remained limited throughout the pandemic. Up until recently, it was the only recommended monoclonal antibody treatment for the Omicron variants, but clinical data are lacking. In a real-world US cohort, among patients unable to take nirmatrelvir/ritonavir, bebtelovimab did not significantly reduce the risk of hospitalization or death (43% reduction, *p* = 0.14) [119]. In a retrospective propensity score study, however, with almost 1000 participants in each arm, bebtelovimab showed a significant reduction in hospitalization by 47% (*p* = 0.01). The benefits were more pronounced among people >65 years old, immunocompromised, and fully vaccinated [120]. Unfortunately, the subvariant BQ.1.1 has demonstrated high resistance against bebtelovimab [87].

Given the growing predominance of BQ.1.1 and the XBB strains that are resistant to bebtelovimab and sotrovimab, the IDSA amended its guidance to recommend against routine use of these monoclonal antibodies in regions where the dominant strains are resistant [121].

### 5.4. Hospital Treatment

#### 5.4.1. Remdesivir

Remdesivir had been originally tested and approved for the treatment of individuals in need of supplemental oxygen and hospitalization. The first positive results from the use of this drug came from the ACTT1 trial [97], wherein 1062 participants were randomized 1:1 to receive either a 5-day course of remdesivir or a placebo for the treatment of COVID-19 necessitating hospital stay. It should be noted that in this study, the primary outcome was not survival, but rather time to recovery, since the study was designed at a point in time when healthcare systems were under extreme pressure, and it was perhaps more important to achieve shortened hospital stays in order to accommodate an increasing number of patients needing admission. Participants in the remdesivir arm had a median length of stay of 10 days, as opposed to 15 days (rate ratio for recovery, 1.29; 95% CI, 1.12 to 1.49; *p* < 0.001, by a log-rank test), and although there was a significant reduction in mortality at 15 days (6.7% vs. 11.9%), this reduction was marginally non-significant at 28 days (11.4% vs. 15.2%, HR 0.73, 95% CI 0.52 to 1.03) [97]. The authors did not mention if any PWH took part in the study.

The choice of softer primary outcome targets in ACTT1 caused some controversy, especially since in the SOLIDARITY trial sponsored by the WHO, the primary endpoint was hospital mortality [121,122]. WHO initially recommended against the use of remdesivir for the treatment of COVID-19 patients in need of hospitalization; this decision was based on the results from the SOLIDARITY trial, where remdesivir, lopinavir/ritonavir, hydroxychloroquine and interferon were tested against the standard of care in an open-label trial. None of the medications showed promising results. In the remdesivir arm in particular, death occurred in 301 of 2743 patients receiving remdesivir and in 303 of 2708 receiving its control (rate ratio, 0.95; 95% CI 0.81 to 1.11, *p* = 0.50). More recently, in a published meta-analysis from the SOLIDARITY group, the authors maintained that remdesivir was not able to reduce mortality and recommended against its use [122]. Again, in the SOLIDARITY trial, there is no mention of participating PWH.

In the most recent iteration of WHO guidelines [123], however, the WHO has a conditional favorable recommendation for remdesivir for patients with serious COVID-19 infection (but not critical) in lieu of data arising from real-world studies. When patients with severe and critical COVID-19 were considered together, pooled analysis demonstrated that remdesivir probably had little or no impact on mortality (OR 0.95, 95% CI 0.84 to 1.07). When considered separately, remdesivir possibly caused an important reduction in mortality (OR 0.89, 95% CI 0.78 to 1.02) in those with severe COVID-19, while possibly having no impact on mortality in those with critical COVID-19 (OR 1.15, 95% CI 0.89 to 1.51). Several real-world studies have been published providing data in favor of remdesivir use, showing a benefit to survival, reduction in mechanical ventilation risk and reduced hospital stay [124,125,126,127].

Remdesivir is not known to have serious drug–drug interactions with ART, and apart from transient hepatotoxicity, it does not seem to have any other side effects of note. There exists a contraindication to using remdesivir in patients under renal replacement therapy due to the accumulation of cyclodextrin. Experience with voriconazole and also several published studies have shown that the brief exposure of 5 days does not warrant these concerns and that remdesivir is safe to use in end-stage renal disease [128,129]. Finally, pertinent to people receiving ART, a recent publication by Shytaz et al. examined the effect of cobicistat on remdesivir treatment and found that cobicistat not only has a moderate antiviral effect on its own, but it can also synergize and enhance remdesivir in an animal model of COVID-19 infection [130].

#### 5.4.2. Dexamethasone

Aside from remdesivir, the other part of mainstay hospital treatment for COVID-19 is the use of corticosteroids, usually dexamethasone. Dexamethasone use in COVID-19 is based mainly on the results of the RECOVERY trial, a large multicenter multifactorial open-label study designed to investigate several treatments for COVID-19 [131]. Patients were randomized 2:1 to receive either dexamethasone or usual care, respectively. Overall, 482 patients (22.9%) in the dexamethasone group and 1110 patients (25.7%) in the usual care group died within 28 days after randomization (age-adjusted rate ratio 0.83, 95%CI 0.75 to 0.93; *p* < 0.001). Of note, dexamethasone did not reduce mortality among patients not requiring supplemental oxygen, but it did reduce mortality among patients under mechanical ventilation. After RECOVERY published its results, dexamethasone was endorsed by most medical associations as a key element of COVID-19 treatment. In the RECOVERY trial, a small number of PWH took part—12 in the dexamethasone arm and 20 in the control group.

The benefits of dexamethasone have been demonstrated in several studies and are summarized in a meta-analysis by the WHO Rapid Evidence Appraisal for COVID-19 Therapies (REACT) Working Group [132]. The researchers found that dexamethasone and hydrocortisone reduced mortality by 33%, whereas the effect of methylprednisolone was more modest, reducing mortality by 10%.

Dexamethasone has several adverse effects, none of which are important for short-term use recommended for COVID-19 treatment. Perhaps more importantly, even though the suggested dose is relatively low, it does pose an added risk for opportunistic infections, mainly PJP and tuberculosis. Healthcare providers need to also account for potential drug–drug interactions when treating PWH for COVID-19 [133]. Theoretically, corticosteroid levels are increased after exposure to ritonavir or cobicistat [73,74], but the short duration and low dose recommended for COVID-19 minimizes any dangers of over-exposure, and co-administration should not be contraindicated. On the other hand, dexamethasone is a CYP3A4 inducer and could reduce therapeutic levels of rilpivirine, and co-administration should be avoided. This is of particular importance in two-drug regimens, where having sub-therapeutic levels of one of them could theoretically lead to the emergence of resistance. Finally, efavirenz is also a CYP3A4 inducer and could lead to reduced levels of dexamethasone, warranting a doubling of the recommended dose [134].

#### 5.4.3. Immunosuppressants

The last step in COVID-19 treatment, particularly in individuals in serious or critical condition, is the use of immunosuppressive drugs apart from corticosteroids. Current guidelines have variably endorsed the use of anakinra, tocilizumab and baricitinib.

Anakinra is a recombinant soluble IL-1 receptor antagonist [135]. There exist several trials investigating its potential use as an adjunctive medication for COVID-19 infection, with conflicting results. One major positive blinded trial from Greece showed that the use of anakinra among patients with suPAR ≥ 6 mg/dl (a surrogate marker of inflammation) cut mortality in half [136]. The generalizability of the results is limited due to the use of a not widely available biomarker, although the authors also suggested that a score using lymphocytes and ferritin levels could be used in its place. On the other hand, a multi-center open-label French study was halted prematurely due to safety concerns after an interim analysis revealed that anakinra use was associated with worse prognosis on day 14 (treatment success 70% vs. 91%, OR 0.23, 95%CI 0.06 to 0.91, *p* = 0.027) and a marginally non-significant increase in 28-day mortality [136]. A Cochrane meta-analysis included four anakinra RCTs and was not able to find any significant effect of anakinra on 28-day mortality or disease progression, albeit with low or very low level of certainty [137]. Nonetheless, anakinra has been recommended for the treatment of COVID-19 infection at a dose of 100 mg in a subcutaneous daily injection for up to 10 days. In its use for COVID-19, it has not been associated with significant side effects, but as with corticosteroid use, one should be alert for opportunistic infections. Finally, there are no significant drug interactions, aside from an additive hematological toxicity when combined with zidovudine [74].

Tocilizumab is a recombinant humanized anti-IL-6 receptor monoclonal antibody that inhibits the binding of IL-6 to both membrane and soluble IL-6 receptors, blocking IL-6 signaling and reducing inflammation [138]. Justification for its use as an adjunctive treatment in COVID-19 came from the RECOVERY trial. In the tocilizumab part of the study, 4116 patients were randomized 1:1 to receive either tocilizumab or standard of care. Overall, 621 (31%) of the 2022 patients who were given tocilizumab and 729 (35%) of the 2094 patients who were given usual care died within 28 days (RR 0.85, 95%CI 0.76 to 0.94, *p* = 0.0028). Tocilizumab was also found to reduce the risk of invasive ventilation by 16%. In this trial, fifteen PWH (seven and eight in each arm) were included. Tocilizumab has received an official indication from the FDA and the EMA for COVID-19 infection.

Several studies since RECOVERY have compared tocilizumab to standard of care, and there have actually been many meta-analyses describing the conflicting results. Godolhin et al. performed a meta-analysis of 19 trials, wherein tocilizumab was found to reduce mortality by 20% [139]. The authors also compared the effects of tocilizumab to sarilumab, another IL-6 receptor antibody, and found no differences in their beneficial effects. Similarly, Piscoya et al., in a meta-analysis of nine RCTs and nine IPTW cohorts, found that tocilizumab reduced all-cause mortality, risk of mechanical ventilation and length of stay [140]. On the other hand, Almeida PRL et al. did not show a benefit to mortality by adding tocilizumab to the standard of care, although it did reduce the risk of mechanical ventilation [141]. Interestingly, there has also been a trial comparing the use of anakinra and tocilizumab, showing a relative risk reduction of 50% and 52% for ICU admission and death, respectively, in favor of anakinra [142].

Tocilizumab does not present significant drug interactions with ART, aside from increased hematological toxicity in combination with zidovudine [74]. Physicians should observe transaminases and neutrophil counts; although, given the one- or two-time administration, one should not expect significant risks. As with other immunosuppressants, there is an increase in the risk of opportunistic infections in PWH, particularly those with low CD4 counts; special attention should be given towards tuberculosis.

Baricitinib is a selective Janus kinase (JAK) 1 and 2 inhibitor that was first investigated as a COVID-19 treatment in the ACTT-2 trial [143]. In ACTT-2, baricitinib was used in combination with remdesivir for the treatment of COVID-19 infection and compared to remdesivir alone. The authors reported 30 participants with immune deficiency, yet there is no mention if any were PWH. Patients receiving high-flow oxygen or noninvasive ventilation at enrollment had a time to recovery of 10 days with combination treatment, and 18 days with control (rate ratio for recovery, 1.51, 95% CI 1.10 to 2.08), but there was no difference in 28-day mortality. Nonetheless, especially at the early stages of the pandemic where it was important to wean patients off of high-flow oxygen devices so they could be used elsewhere, this benefit was very important. The problem with ACTT-2 was that when it was designed and conducted, dexamethasone had not yet been shown to be beneficial, and thus it was not part of the standard of care at the time, making the results from this trial difficult to interpret after the publication of RECOVERY.

COV-BARRIER would come to bridge the gap in literature by investigating the use of baricitinib in critically ill patients, in addition to remdesivir and dexamethasone [144]. Treatment with baricitinib significantly reduced 28-day all-cause mortality compared to placebo (39% of participants died in the baricitinib group vs. 58% in the placebo group; HR 0.54, 95%CI 0.31 to 0.96], *p* = 0.030; 46% relative reduction; absolute risk reduction 19%). As with the other immunosuppressants, meta-analyses have been published for baricitinib as well. Manoharan et al. included 15 trials in their study and showed that baricitinib in addition to standard of care could reduce mortality by 40% [145]. Cherian et al. went a step further and compared baricitinib to tocilizumab in their meta-analysis, and they reported that treatment with baricitinib (RR 0.69, 95%CI, 0.50 to 0.94, *p* = 0.02) but not with tocilizumab (RR 0.87, 95%CI 0.71 to 1.07, *p* = 0.19) reduced 28-day mortality [146].

Baricitinib is not expected to have any important drug interactions with ART, apart from the risk of increased hematological toxicity when combined with zidovudine [74]. As with the other immunosuppressants, it can increase the risk of opportunistic infections, with a special warning for tuberculosis. It has also been associated with an increased risk of thrombosis, although this risk was not recorded in the COVID-19 trials. Nonetheless, physicians should maintain an increased alertness for thromboses, especially since COVID-19 infection itself has also been implicated in immunothrombosis.

## 6. Lessons Learned and Future Directions

The COVID-19 pandemic disrupted HIV prevention and treatment programs, and it threatened the global response to the HIV/AIDS pandemic. Implementing measures to stop the new virus from spreading had important implications for maintaining health services related to HIV in many parts of the world. Emerging evidence suggests that people from vulnerable populations and racial or gender minorities had experienced high morbidity and mortality related to COVID-19, widening the existing inequalities among HIV key populations. PWH that reside in low-income countries experienced additional barriers that inhibited access to care and maintenance of their antiviral treatment medication [147]. Moreover, geographic disparities existed for the coverage of COVID-19 vaccination as well as for antivirals for COVID-19 treatment. According to the WHO, the overall median completed primary series COVID-19 vaccination coverage was 59%, ranging from 21% (low-income countries) to a high of 74% (high-income countries) [148]. In addition, due to the high cost, antivirals and monoclonal antibodies have limited availability in developed countries.

Unfortunately, the disruption of HIV services may persist for a long time due to the ongoing global economic crisis. According to a UNAIDS report, data have revealed a worrisome situation. Eastern Europe, Central Asia, the Middle East, North Africa and Latin America have reported increases in annual HIV infections over the past decade. Moreover, in Asia and the Pacific, new HIV infections are rising where they had been falling over the past 10 years. Malaysia and the Philippines are among the countries with rising epidemics among key populations [149]. The WHO highlights the importance of ensuring continuous access to HIV services in all settings. In addition to fear and ignorance for the new disease, many forms of stigma and discrimination have surfaced. In several countries, lockdowns have hampered the opportunity to get tested and promptly treated for HIV [150].

The recent pandemic has shown us that anyone can be infected with the new virus. It is of the utmost importance to eliminate both existing HIV and COVID-19-related stigmas and ensure that PWH are not further marginalized. This must be an integral component of the global efforts to respond to the pandemic [151].

Moreover, PWH ignorant of their HIV status have an increased risk of COVID-19 and related complications. Some of them may have comorbidities (e.g., diabetes, cardiovascular disease, chronic kidney, liver disease, lung disease) that are known risk factors for the complications of COVID-19. Given the reality that medical services were overburdened due to COVID-19, HIV Self Testing (HIVST) in the COVID-19 context has many advantages. HIVST can play a critical role in ensuring the continuity of HIV testing services during this time, as it provides an opportunity to reach people who may not otherwise get tested and reduce the number of people attending medical facilities. HIVST nowadays can be easily adopted, and countries with regulatory barriers should enable its implementation and widespread access to overcome any obstacle may exist [152,153,154].

Vaccines preventing SARS-CoV-2 infection changed the evolution of the COVID-19 epidemic as they substantially reduced the risk of COVID-19 and have been associated with reductions in COVID-19-associated hospitalizations and deaths. However, it has recently been reported that following SARS-CoV-2 mRNA vaccination, a transient HIV expression occurs, manifesting primarily as the activation of HIV-Nef-specific CD8+ T-cell responses. Even though researchers have not observed a significant depletion of intact proviruses, this CD8+ T-cell induction is correlated with significant decreases in cell-associated HIV mRNA, suggesting the killing or suppression of cells transcribing HIV [155]. Some mRNA vaccines targeting HIV antigens are under development for both prophylactic and therapeutic settings. Future studies may enable reservoir reductions by mRNA vaccines encoding HIV antigens [156,157].

## 7. Conclusions

COVID-19 and HIV are reminiscent of Plutarch’s Parallel Lives—two different diseases associated with similar media frenzy, fear of the unknown and stigma. The COVID-19 presentation and risk factors in PWH are similar to that of the general population, and among virally suppressed PWH, the outcomes are not much different. If anything, this is a testament to what modern medicine has achieved for PWH, essentially minimizing their increased risk for unfavorable outcomes through the use of modern ART. Physicians need to maintain awareness of certain interactions of ART and COVID-19 treatments, but even more, they need to be up to date of currently circulating strains in an ever-changing sea of variants and available monoclonal antibody treatments. Our hope remains that both pandemics will, in the future, only be part of literature—similar to that of Plutarch—and be extinct from every-day life.

## Figures and Tables

**Figure 1 viruses-15-00577-f001:**
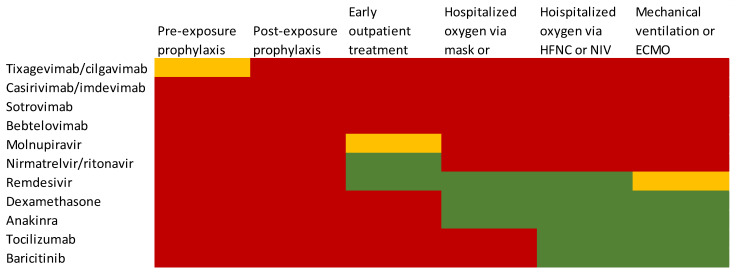
Current guidelines for the treatment of COVID-19. Red: this treatment is no longer recommended. Orange: this treatment is recommended lacking other options or in specific regions and populations. Green: this treatment is recommended for all patients.

## Data Availability

Not applicable.

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
