# Peer review of "HIV and COVID-19 Co-Infection: Epidemiology, Clinical Characteristics, and Treatment"

_viruses, 2023, doi:10.3390/v15020577_

Round 1

Reviewer 1 Report

This is a very comprehensive review with an abundance of supporting data. The disparities regarding HIV and COVID were brought up and how that may affect the epidemiology, perhaps a few lines regarding why may be helpful.  Access to vaccines and access to medicine I believe are great issues especially for epidemiology of diseases and though there were comments regarding this, a few additional connections may be warranted.  This, I believe, would be optional

Author Response

Thank you for your insightful comments. We have added a small paragraph at the end of our manuscript discussing in brief the problem of unequal access to care.   

Reviewer 2 Report

This review described the epidemiology, clinical characteristic and managing HIV and COVID-19 co-infection. The topic was interesting and well-written. However, there were imbalanced contents between the subtopics of the epidemiology, the clinical characteristics, and the management/treatment. There were eight pages on the management/treatment, two for epidemiology and only one for clinical characteristics. Therefore, I like to recommend the authors elaborate and update the information based on the comments below:

1. I suggest including the prevalence/incidence of HIV among COVID-19 patients and the disease burden.

2. Line 71-74: “Regarding the likelihood of hospitalization due to COVID-19 among PWH…..” The authors may update the information based on this current systematic review:

Danwang, C., Noubiap, J.J., Robert, A. et al. Outcomes of patients with HIV and COVID-19 co-infection: a systematic review and meta-analysis. AIDS Res Ther 19, 3 (2022). https://doi.org/10.1186/s12981-021-00427-y

3. Line 104-106: “Tuberculosis co-infection was associated with an increased hazard of COVID-19…….” Please rephrase these statements. The statement seems unrelated to the topic.

4. I suggest adding a subtopic of clinical outcomes and separating it from the subtopic epidemiology so that readers have a clearer view of these subtopics.

5. I recommend the authors elaborate more and update on HIV and COVID-19 vaccination.

Author Response

We appreciate your help in improving our work. There is an imbalance in the size of the manuscript's sections, but there is a lot more to be said on treatment rather than epidemiology, outcomes and clinical characteristics, therefore, we feel, this is inevitable. There are a lot of treatments that have been tried and abandoned and we opted to not even expand on those (ie hydroxychloroquine or inhaled budesonide or fluvoxamine etc) and limit ourselves to what we believe is relevant to the current phase of the pandemic or might become relevant in the future (which is why we include discussion on the use of monoclonal antibodies, even though currently none is actually recommended any more). For your other comments:  1) Indeed, great omission on our behalf not to at least include some numbers on incidence and prevalence. Thank you very much for the input. We have added a paragraph in the epidemiology section (in yellow) 2) The suggested work has been added and cited within the review in the epidemiology part 3) We believe that the statements on HCV and tuberculosis are relevant to the reader, since they represent particular risk factors associated with HIV, either because TB is a frequent opportunistic infection or because there is an overlap with HCV and HIV pandemics in populations of people who inject drugs. We did, however, include it in a separate paragraph and added a sentence to note why we believe this information is relevant to this review. 4) We separated the subtopics per your suggestion 5) When we decided on this review, we debated whether we should include a review on vaccination and decided that it would be beyond our scope. Nonetheless at the beginning of the management section (in yellow) we have added a small paragraph both mentioning that our review will not elaborate on vaccines, but also discussing in brief what is known about vaccination in PWH against COVID-19. There is also a small part at the end of our manuscript, related to input from the other reviewer, concerning inequality of access to vaccines and medications.